## [Decision Letter · Decision Letter 0]

16 Sep 2021

PONE-D-21-10417

ARL11 correlates with the immunosuppression and poor prognosis in breast cancer: a comprehensive bioinformatics analysis of ARL family members

PLOS ONE

Dear Dr. Song,

Thank you for submitting your manuscript to PLOS ONE. After careful consideration, we feel that it has merit but does not fully meet PLOS ONE’s publication criteria as it currently stands. Therefore, we invite you to submit a revised version of the manuscript that addresses the points raised during the review process.

We look forward to receiving your revised manuscript.

Kind regards,

Chandi C. Mandal, Ph.D.

Academic Editor

PLOS ONE

Additional Editor Comments (if provided):

Your manuscript has been evaluated by two independent subject experts. We have received comments from both reviewers. As per reviewers' views and editor opinion, this manuscript can not be considered for its acceptance. However, you may submit revised version after incorporating/addressing all queries raised by reviewers. Reviewers comments are given below.

Journal Requirements:

3. Thank you for stating the following financial disclosure: "NO"

4. Thank you for stating the following in your Competing Interests section: "NO"

7. We noticed you have some minor occurrence of overlapping text with the following previous publication, which needs to be addressed:

- https://www.researchsquare.com/article/rs-35982/v1

In your revision ensure you cite all your sources (including your own works), and quote or rephrase any duplicated text outside the methods section. Further consideration is dependent on these concerns being addressed.

Reviewers' comments:

Reviewer's Responses to Questions

5. Review Comments to the Author

Reviewer #1: 1. Figures are of poor quality. In several figures, including figure 1 (A-C), Labels on X-Axis and Y-Axis are not legible. Quality of Figures should be improved.

2. Figures have not been arranged properly.

3. Results- 1st paragraph 3rd line, authors have first mentioned S3 Fig. and then S1 Fig. Authors should re-arrange the supplementary Figures in the sequential order as they appear in the text.

4. In S3 Fig A., gene name should be aligned properly with the rows. Coding for the red, blue and black lines shall be explained. Figure S3A Fig, should be moved up.

5. In S3 Fig B., interpretation for all the colored lines shall be explained. FS3A and S3B Figures should be split into two separate figures. Type of mutations found should also be explained, especially with ARL8A.

6. Results- 1st paragraph 5th line, the word ‘obviously’ should be replaced with ‘significantly’

7. Authors have mentioned that ARL11 significantly correlate with infiltration of CD8+ T cells (R = 0.062, P = 0.05). Correlation coefficient of 0.062 is not practically relevant. Therefore, authors should not use describe they found a strong positive correlation of ARL11 with CD8+ cells.

8. Immune infiltration data should be analyzed with TIMER 2.0 which employs 6 different algorithms, and also it gives analysis for different Th subtypes and macrophage subtypes.

Reviewer #2: The manuscript describes research that analyzes ADP-ribosylation factor-like protein (ARL) family members in breast cancer. The research is based on bioinformatic analyses. The study concludes that of the ARLs, ARL11 is the most significant prognostic indicator in breast cancer. Its expression correlates with relapse-free survival as well as genes in immunogenic pathways that suppress cancer progression.

The weakness of the study is that the same breast cancer cohort does not underlie all the analyses that are performed. The analyses are performed by bioinformatic tools on public websites and thus a certain control of the data is lost, control such as normalization of the data, which ARL transcript was used, etc. Still, the results of the study are intriguing and ARL11 may be a potential new target in breast cancer.

Below are some points that the authors are asked to address.

Major points:

The authors should show Kaplan-Meier survival data from a TCGA breast cancer cohort, i.e., the same dataset that is used to analyze differences in expression between normal breast tissue and tumors and that is the basis for most other analyses in the study. The Kaplan-Meier (KM) plotter is a great tool for exploration but it relies on various datasets run at different centers and it does not include breast cancer data from TCGA. Is it possible that the discrepancy in the study that is observed between expression data and survival are dependent on different types of datasets used? E.g., expression of ARL2 and ARL9 is lower in breast tumors than normal breast tissue but in the survival analyses it is higher expression levels of the two genes that correlate with shorter survival. This result is opposite to what is expected and it is not in line with ARL11 results, i.e. higher expression in tumor than normal and higher expression correlates with shorter survival. Is this discrepancy due to different datasets used to perform different sets of experiments?

Since the study is solely based on bioinformatic analyses, it is necessary to confirm the main findings, i.e. ARL11 in relapse-free survival and its effect on immunosuppression, in another large breast cancer cohort. Perhaps the KM plotter could be used for that purpose but the authors should make clear that it is a combination of different datasets. METABRIC (n = 2509) is available in cBioPortal with clinical and expression data.

It is not clear from the study how the two TCGA cohorts were used in the study. Since there is overlap between patients in the two TCGA cohorts used in the study, they can not be used to confirm results obtained in one another. Please clarify in the manuscript which cohort was used for what analysis.

Minor points:

Please include in both Methods and figure legends what breast cancer cohorts are used in each bioinformatic analyses (Methods) and the results shown (figures/figure legends). TCGA datasets are measured on the same platforms and therefore they can be used by different analytical tools and the results compared. Using datasets that are measured on different platforms can affect the outcome of statistical and bioinformatical analyses.

Figure legend 2. Please include the color code key that is currently missing. Is red = amplification, etc?

Figure legend 4. Please include how many patients are in each group, high and low.

Figure 6A. The figures generated by TIMER clearly state “partial cor.” Can the authors please clarify what partial means in the context of the analysis. Is the correlation only complete, or noteworthy, once the coefficient rises above a certain threshold?

It would benefit the study to include figure legends for the Supporting figures.

The numbering of Supporting figures in the manuscript should be re-arranged. The first Supporting figure cited starts with S3A Fig rather than S1 Fig.

Words like “remarkably” and “obviously” are subjective and are best left out (Results section).

Sub-chapter entitled ”Genetic alterations of ALRs in BC:” there is a reference to S2B Fig that I think is supposed to be S3B Fig. Please check. In the same chapter, it says that there “was a negative correlation between mRNA expression and DNA methylation of most ARLs”. Please include an explanation in the text why ARL4A, 4C and 11 were singled out to show in the manuscript.

Sub-chapter called “The prognostic values of ARLs in BC.” …………”whereas higher expression levels of ARL15….” should read …”whereas LOWER expression.” Later in the same sub-chapter it says “ARL11 was the most significant prognostic indicator…” Can you please explain in the manuscript why this conclusion was drawn. There are other interesting ARLs in the study and some with opposing effects to ARL11.

Immune infiltration in breast cancer is subtype-dependent and the results shown in S3 Table support that. The results for breast cancer are interesting and warrant mention in the Results chapter where ARL11 levels are correlated with immune suppression. Please include a sentence or two.

Discussion

Multiple sites in ARL4A and 4C are hypomethylated in breast tumors and yet there is lower expression of these family members in tumors than normal breast tissue. One would expect higher expression from a gene that is hypomethylated.

Please clarify.

In Results section the authors suggest that “ARL11 copy number amplification……..involved in epigenetic regulation.” In Discussion section it says that ARL11, located at 13q14.3, “is frequently deleted ……” These sentences oppose one another. According to the data presented in the manuscript one would expect amplification of ARL11 in breast tumors. What is the status of the copy number of ARL11 in the breast tumors?

The results in the manuscript are interesting but they are still preliminary and the authors should mention that more work needs to be done to prove their hypothesis that ARL11 is prognostic in BC.

---

## [Author Response · Author response to Decision Letter 0]

20 Feb 2022

Dear editors,

We firstly thank all the reviewers for their thorough reading of our manuscript and for the constructive suggestions/criticisms. Our point-by-point responses and the corresponding changes to the manuscript are described below.

Reviewer #1:

1. Figures are of poor quality. In several figures, including figure 1 (A-C), Labels on X-Axis and Y-Axis are not legible. Quality of Figures should be improved.

Response: Thanks for your careful review. We have improved the quality of figures.

2. Figures have not been arranged properly.

Response: Thank you. We have rearranged the images in figures.

3. Results- 1st paragraph 3rd line, authors have first mentioned S3 Fig. and then S1 Fig. Authors should rearrange the supplementary Figures in the sequential order as they appear in the text.

Response: Thanks for your suggestion. We have rearranged the supplementary Figures in the sequential order as they appear in the text.

4. In S3 Fig A., gene name should be aligned properly with the rows. Coding for the red, blue and black lines shall be explained. Figure S3A, should be moved up.

Response: In line with the reviewer’s suggestion, we gave more detailed information on ARLs (ADP-ribosylation factor-like subfamily members) in the Introduction section.

5. In S3 Fig B., interpretation for all the colored lines shall be explained. FS3A and S3B Figures should be split into two separate figures. Type of mutations found should also be explained, especially with ARL8A.

Response: Thank you. We have explained all the colored lines and mutation types in Figure legends for S3 Fig.

6. Results- 1st paragraph 5th line, the word ‘obviously’ should be replaced with ‘significantly’.

Response: Thanks for your careful review. We have replaced this word in 1st paragraph.

7. Authors have mentioned that ARL11 significantly correlate with infiltration of CD8+ T cells (R = 0.062, P = 0.05). Correlation coefficient of 0.062 is not practically relevant. Therefore, authors should not use describe they found a strong positive correlation of ARL11 with CD8+ cells.

Response: Thanks for your great advice. TIMER 2.0 is a great tool to comprehensively analyze immune infiltrates across diverse cancer types. Therefore, we reanalyzed the correlation between ARL11 and immune infiltrates in BRCA in our revised study. Additionally, we corrected our inappropriate description in manuscript. 

8. Immune infiltration data should be analyzed with TIMER 2.0 which employs 6 different algorithms, and also it gives analysis for different Th subtypes and macrophage subtypes.

Response: Thanks for your suggestion. we reanalyzed the correlation between ARL11 and different Th subtypes as well as macrophage subtypes in BRCA in our revised version of manuscript by TIMER 2.0 database.

Reviewer #2: The manuscript describes research that analyzes ADP-ribosylation factor-like protein (ARL) family members in breast cancer. The research is based on bioinformatic analyses. The study concludes that of the ARLs, ARL11 is the most significant prognostic indicator in breast cancer. Its expression correlates with relapse-free survival as well as genes in immunogenic pathways that suppress cancer progression.

The weakness of the study is that the same breast cancer cohort does not underlie all the analyses that are performed. The analyses are performed by bioinformatic tools on public websites and thus a certain control of the data is lost, control such as normalization of the data, which ARL transcript was used, etc. Still, the results of the study are intriguing and ARL11 may be a potential new target in breast cancer. Below are some points that the authors are asked to address.

Major points:

1. The authors should show Kaplan-Meier survival data from a TCGA breast cancer cohort, i.e., the same dataset that is used to analyze differences in expression between normal breast tissue and tumors and that is the basis for most other analyses in the study. The Kaplan-Meier (KM) plotter is a great tool for exploration but it relies on various datasets run at different centers and it does not include breast cancer data from TCGA. Is it possible that the discrepancy in the study that is observed between expression data and survival are dependent on different types of datasets used? E.g., expression of ARL2 and ARL9 is lower in breast tumors than normal breast tissue but in the survival analyses it is higher expression levels of the two genes that correlate with shorter survival. This result is opposite to what is expected and it is not in line with ARL11 results, i.e. higher expression in tumor than normal and higher expression correlates with shorter survival. Is this discrepancy due to different datasets used to perform different sets of experiments?

Response: Thanks for your valuable suggestion. 

1. To avoid the discrepancy among different datasets, we also reanalyzed the expression level of ARLs in tumor and normal tissues as well as the clinical outcome (overall survival) for BC patients with different ARLs expression by TIMER 2.0 database. 

TIMER 2.0 (http://timer.comp-genomics.org/) is a comprehensive and effective tool to explore tumor immunological, clinical and genomic features based on TCGA dataset, and allows users to generate high-quality figures. The expression data of ARLs from TIMER 2.0 are consistent with the result which we have shown before (Figure 1 and S1 Table). Additionally, the survival analysis models showed that only ARL11 was significantly correlated with the poor clinical outcome for BC patients among ARL family members (Supporting information).

2. Also, we observed the contradiction between expression data and survival data in TCGA database. For example, ARL2 is significantly downregulated in BC tumor tissues compared with normal tissues (P < 0.01). Conversely, we found the obvious tendency that higher expression of ARL2 might indicate poor overall survival for BC patients (p > 0.05). We have observed the same results for ARL6 and ARL13A, which has been shown below. For those genes, their dysregulation might function in the early stage of BC and be associated with the occurrence of BC. However, they would lose their anti/pro-cancer effect in BC progression. On the other hand, tumor development and progression are complex and variable, this gene does not act alone, or might be passively regulated by other oncogenes and tumor-suppressor factors. Meanwhile, we have added this statement in discussion part in revised version of manuscript.

2. Since the study is solely based on bioinformatic analyses, it is necessary to confirm the main findings, i.e. ARL11 in relapse-free survival and its effect on immunosuppression, in another large breast cancer cohort. Perhaps the KM plotter could be used for that purpose but the authors should make clear that it is a combination of different datasets. METABRIC (n = 2509) is available in cBioPortal with clinical and expression data. It is not clear from the study how the two TCGA cohorts were used in the study. Since there is overlap between patients in the two TCGA cohorts used in the study, they can not be used to confirm results obtained in one another. Please clarify in the manuscript which cohort was used for what analysis.

Response: 

1. Thanks for your valuable suggestion. METABRIC (n = 2509) is a comprehensive resource for clinical analysis, but it is regrettable that we have not obtained valid information for the clinical relevance of immune infiltrates and ARL11 in METABRIC dataset. In revised manuscript, we reanalyzed the correlation between ARL11 and different subtypes of immune infiltrates in BRCA. Additionally, we corrected our inappropriate description in manuscript. We conducted multivariable Cox proportional hazard model to explore the clinical relevance of tumor immune infiltrates as well as ARL11 expression through TIMER 2.0 database. 

2. For correlation analysis and genetic alteration analysis, we unified to use the same breast cancer dataset (TCGA, Firehose Legacy, n = 1108) in cBioPortal. We have corrected the corresponding parts in revised manuscript.

Minor points: 

1. Please include in both Methods and figure legends what breast cancer cohorts are used in each bioinformatic analyses (Methods) and the results shown (figures/figure legends). TCGA datasets are measured on the same platforms and therefore they can be used by different analytical tools and the results compared. Using datasets that are measured on different platforms can affect the outcome of statistical and bioinformatical analyses. 

Response: Thanks. We have corrected this point in revised manuscript.

2. Figure legend 2. Please include the color code key that is currently missing. Is red = amplification, etc?

Response: Thanks for your suggestion, we have added the annotation for each color code in revised Figure 2 and figure legend.

3. Figure legend 4. Please include how many patients are in each group, high and low.

Response: Thank you. We have added it in revised manuscript.

4. Figure 6A. The figures generated by TIMER clearly state “partial cor.” Can the authors please clarify what partial means in the context of the analysis. Is the correlation only complete, or noteworthy, once the coefficient rises above a certain threshold?

Response: Thanks for your advice. We have corrected this inappropriate statement in revised version.

5. It would benefit the study to include figure legends for the Supporting figures.

Response: Thank you. We have added the details in figure legends for the Supporting figures.

6. The numbering of Supporting figures in the manuscript should be re-arranged. The first Supporting figure cited starts with S3A Fig rather than S1 Fig.

Response: Thanks. We have rearranged the supplementary Figures in the sequential order as they appear in the text.

7. Words like “remarkably” and “obviously” are subjective and are best left out (Results section).

Response: Thanks for your advice. We have replaced such words in Results section.

8. Sub-chapter entitled “Genetic alterations of ARLs in BC:” there is a reference to S2B Fig that I think is supposed to be S3B Fig. Please check. In the same chapter, it says that there “was a negative correlation between mRNA expression and DNA methylation of most ARLs”. Please include an explanation in the text why ARL4A, 4C and 11 were singled out to show in the manuscript.

Response: Thank you.

1. We have corrected the mistakes in figure order.

2. Additionally, in revised manuscript, we reanalyzed the genetic alteration of ARLs in breast cancer dataset (TCGA, Firehose Legacy, n = 1108). We found that there was a negative correlation between mRNA expression and DNA methylation of most ARLs. As we mentioned in Result section, ARL3, ARL4C, 4D and 11 were selected for their higher correlation coefficient among all ARLs (R > 0.5 and P < 0.01). we have showed the detailed data of DNA methylation status of ARLs in S3 Table.

9. Sub-chapter called “The prognostic values of ARLs in BC.” …………” whereas higher expression levels of ARL15….” should read …” whereas LOWER expression.” Later in the same sub-chapter it says “ARL11 was the most significant prognostic indicator…” Can you please explain in the manuscript why this conclusion was drawn. There are other interesting ARLs in the study and some with opposing effects to ARL11. 

Response: Thanks for your careful review. ARL11 was selected as the most significant clinical indicator among ARLs for BC patients after comprehensively evaluating their expression status and prognostic values. We have added the sentences in Result section.

10. Immune infiltration in breast cancer is subtype-dependent and the results shown in S3 Table support that. The results for breast cancer are interesting and warrant mention in the Results chapter where ARL11 levels are correlated with immune suppression. Please include a sentence or two.

Response: Thanks for your valuable suggestion. We have added the sentences into Result section.

Discussion

1. Multiple sites in ARL4A and 4C are hypomethylated in breast tumors and yet there is lower expression of these family members in tumors than normal breast tissue. One would expect higher expression from a gene that is hypomethylated. Please clarify.

Response: Thanks. 

DNA methylation is critical to regulate gene expression. However, in tumor progression, the mRNA or protein expression for gene are affected by several dynamic factors, such as cancer hallmark pathways and regulation by transcription factors (enhancer). Our previous study also demonstrated that TGF-B1 stimulation would significantly upregulate the protein expression of ARL4C in gastric cancer cells. Although ARL4C genes are found to be hypomethylated in BC, its expression might be also regulated by TGF-B1 signal or other cancer hallmark pathways. In line with your valuable suggestion, we have added this statement into Discussion section in revised manuscript.

2. In Results section the authors suggest that “ARL11 copy number amplification……..involved in epigenetic regulation.” In Discussion section it says that ARL11, located at 13q14.3, “is frequently deleted ……” These sentences oppose one another. According to the data presented in the manuscript one would expect amplification of ARL11 in breast tumors. What is the status of the copy number of ARL11 in the breast tumors? 

The results in the manuscript are interesting but they are still preliminary and the authors should mention that more work needs to be done to prove their hypothesis that ARL11 is prognostic in BC.

Response: Sorry for this misleading statement. Our results proved that copy number amplification and DNA methylation might contribute to the epigenetic regulation of most ARLs. Before we finished the prognostic analysis, we did not confirm which ARLs was the most significant clinical factors for BC patients. So, we did not focus on the alteration types of ARL11 in BC.

To evaluate the CNA status ARL11 in BC, we reanalyzed its genetic alteration in 7 independent BC datasets in cBioportal. The result indicated that there are 3 mutation types for ARL11 in BC and vary across different BC subtypes. As shown in the following figure, deep deletion is mainly existed in Breast Mixed Ductal and Lobular Carcinoma. Both deep deletion and CAN are existed in Breast Invasive Lobular Carcinoma and Breast Invasive Ductal Carcinoma, and missense mutation mainly occurs in Invasive Breast Carcinoma.

Finally, we thank again you and the reviewers for your great efforts on improving the quality of this manuscript. We are looking forward to hearing your decision soon.

Thank you very much! 

Best wishes,

Yours sincerely,

Yahua Song

---

## [Decision Letter · Decision Letter 1]

11 Apr 2022

PONE-D-21-10417R1ARL11 correlates with the immunosuppression and poor prognosis in breast cancer: a comprehensive bioinformatics analysis of ARL family membersPLOS ONE

Dear Dr. Song,

Thank you for submitting your manuscript to PLOS ONE. After careful consideration, we feel that it has merit but does not fully meet PLOS ONE’s publication criteria as it currently stands. Therefore, we invite you to submit a revised version of the manuscript that addresses the points raised during the review process.

Your revised manuscript has again been evaluated by the same reviewers. However, both of them have raised various concerns which are to be addressed.

We look forward to receiving your revised manuscript.

Kind regards,

Chandi C. Mandal, Ph.D.

Academic Editor

PLOS ONE

Reviewers' comments:

Reviewer's Responses to Questions

6. Review Comments to the Author

Reviewer #1: The revised manuscript is better than the previous version, however there are several concerns that need to addressed.

1. Introduction: Last Paragraph, the First sentence, “To date, we initially synthetically, the use of word ‘synthetically’ is not appropriate, because authors have not synthesized anything.

2. Methods Section: First Paragraph, the First sentence, “The use of synonym ‘Synthetical and co-active’ for bioinformatic tools is also not appropriate.

3. Results: first paragraph, 4th sentence: Authors needs to confirm whether normal samples used in the analysis are ‘adjacent’ tissues from cancer patients or are they taken from normal patients with benign disease

4. Sub-heading of Results- Genetic alterations of ARLs in BC “In order to synthetically seize the expression profiles……”. Here use of the words “synthetically seize” are not appropriate.

5. Sub-heading of Results- The prognostic values of ARLs in BC “We ulteriorly explored……”. Here use of the word “ulteriorly” is not appropriate.

6. In Figure 6 A, Authors have shown association of ARL11 expression with immune infiltration. “As shown in Fig 6A, we discovered that high ARL11 expression positively involved in the infiltration levels of CD4+ T cells (R = 0.084, P = 8.10E-03), B cells (R = 0.127, P = 6.04E-05), regulatory T cells (Tregs) (R = 0.103, P = 1.13E-03), neutrophil cells (R = 0.072, P = 2.36E-02) and M2 tumor-associated macrophages (TAMs) (R = 0.119, P = 1.67E-04). While, high ARL11 expression negatively associated with the infiltration levels of dendritic cells (DCs) (R = -0.077, P = 1.56E- 02).” In this analysis correlation coefficient is less than 0.2. Any correlation coefficient less than +0.2 or -0.2 is very weak and I mention previously these kinds of statistical associations have no practical relevance. Therefore, this paragraph and figure should be removed from the manuscript.

7. Quality of Figures is better than previous version, however the Figures are still not of publication quality. For example, Panels of Figure 5A, %C and 5D are not legible. Labels on X-axis for Figure 4A are also not legible.

8. Several non-scientific words such as “ecumenically”, “Synthetical and co-active”, synthetically seize “Ulteriorly snatch, “scavenge the mechanism” etc have been used throughout the manuscript. Therefore, the manuscript needs to be proofread by a professional service provider.

Reviewer #2: The answers to the comments made by previous reviewers should be incorporated into the manuscript. It has been done in most cases but not all cases. If a result is not replicated in another cohort as requested by reviewer #2, rather than omitting that information from the manuscript, it can be briefly mentioned, explained and discussed in discussions. The overall survival in TIMER2.0 is not a confirmation of the relapse-free survival in KM plotter, because overall survival and relapse-free survival are not the same.

The conclusions from figure 6 should be toned down (Fig6A) and they appear incorrect in Fig6B (Fig6B Macrophage M2). First, even though immune infiltrates significantly correlate with ARL11 expression, the R values in Fig 6A are close to zero, and therefore an increase in ARL11 expression would result in a very small increase of immune infiltrates. Second, the data in Fig6B do not support that high ARL11 associates with shorter survival, they only support that high M2-TAMs associate with shorter survival. The significant effect is only due to the level of high M2-TAMs, which can be seen in the comparison shown in the figure between: 2 vs 1, which compares low ARL11 + high M2 (light blue line) with low ARL11 + low M2 (dark blue line), and 4 vs 3, which compares high ARL11 + high M2 (red line) with high ARL11 + low M2 (yellow line).

The expression values of the ARLs were evaluated in TIMER 2.0. Please include a sentence to inform the reader whether the results were the same in UALCAN and TIMER, i.e. higher in tumors than normal or vice versa.

According to the manuscript, the methylation status of ARL4A is supposed to be seen in Table 1, but there is only ARL4C and ARL4D.

„…..copy number amplification and DNA methylation might be involved in the epigenetic regulation of ARLs.“ Copy number amplification (CNA) is not considered an epigenetic mechanism. However, CNA may well regulate ARL expression levels.

The figure legends can be more detailed in many cases. For example, Fig 3 is completely missing a legend, Fig 4 would be easier to understand if more explanations were included, and in Fig 6 there is not a positive correlation with CD8+ T cells or M1-TAMs. Please amend.

They are called Circos plots with ´o´ not Circus with ´u´.

The text has been changed for the worse in the tracked version of the manuscript. The original text was more appropriate in many instances. Please fix.

---

## [Author Response · Author response to Decision Letter 1]

20 Jun 2022

Dear editors,

We firstly thank all the reviewers for their thorough reading of our manuscript and for the constructive suggestions/criticisms. Our point-by-point responses and the corresponding changes to the manuscript are described below.

Reviewer #1:

1. Introduction: Last Paragraph, the First sentence, “To date, we initially synthetically, the use of word ‘synthetically’ is not appropriate, because authors have not synthesized anything.

Response: Thanks for your careful review. We have replaced this word in last paragraph, Introduction section.

2. Methods Section: First Paragraph, the First sentence, “The use of synonym ‘Synthetical and co-active’ for bioinformatic tools is also not appropriate.

Response: Thank you. We have corrected this inappropriate sentence into “UALCAN is a comprehensive, user-friendly, and interactive web-portal”.

3. Results: first paragraph, 4th sentence: Authors needs to confirm whether normal samples used in the analysis are ‘adjacent’ tissues from cancer patients or are they taken from normal patients with benign disease.

Response: Thanks for your careful review. The description of “adjacent normal tissues” is not accurate and we have corrected this error in revised manuscript.

4. Sub-heading of Results- Genetic alterations of ARLs in BC “In order to synthetically seize the expression profiles……”. Here use of the words “synthetically seize” are not appropriate.

Response: Thanks for your great advice. We have changed those inappropriate statements in revised version of manuscript.

5. Sub-heading of Results- The prognostic values of ARLs in BC “We ulteriorly explored……”. Here use of the word “ulteriorly” is not appropriate.

Response: Thank you. We have corrected “ulteriorly” into “furtherly”.

6. In Figure 6 A, Authors have shown association of ARL11 expression with immune infiltration. “As shown in Fig 6A, we discovered that high ARL11 expression positively involved in the infiltration levels of CD4+ T cells (R = 0.084, P = 8.10E-03), B cells (R = 0.127, P = 6.04E-05), regulatory T cells (Tregs) (R = 0.103, P = 1.13E-03), neutrophil cells (R = 0.072, P = 2.36E-02) and M2 tumor-associated macrophages (TAMs) (R = 0.119, P = 1.67E-04). While, high ARL11 expression negatively associated with the infiltration levels of dendritic cells (DCs) (R = -0.077, P = 1.56E- 02).” In this analysis correlation coefficient is less than 0.2. Any correlation coefficient less than +0.2 or -0.2 is very weak and I mention previously these kinds of statistical associations have no practical relevance. Therefore, this paragraph and figure should be removed from the manuscript.

Response: Thanks for your suggestion. We have removed this paragraph and related-figure in the revised manuscript.

7. Quality of Figures is better than previous version, however the Figures are still not of publication quality. For example, Panels of Figure 5A, %C and 5D are not legible. Labels on X-axis for Figure 4A are also not legible.

Response: Thank you very much! We have improved the quality of figures in revised version of manuscript.

8. Several non-scientific words such as “ecumenically”, “Synthetical and co-active”, synthetically seize “Ulteriorly snatch”, “scavenge the mechanism” etc. have been used throughout the manuscript. Therefore, the manuscript needs to be proofread by a professional service provider.

Response: Thanks for your careful review. We have corrected those inappropriate statements in revised version of manuscript.

Reviewer #2: 

1. The overall survival in TIMER2.0 is not a confirmation of the relapse-free survival in KM plotter, because overall survival and relapse-free survival are not the same.

Response: Thanks for your great advice. In line with your valuable opinion, the overall survival analysis in TIMER2.0 is not a confirmation for Relapse-free survival in KM plotter in our study. We performed OS analysis to comprehensively explore the clinical values and found only ARL11 had significant prognostic values in both TCGA and GEO datasets.

2. The conclusions from figure 6 should be toned down (Fig6A) and they appear incorrect in Fig6B (Fig6B Macrophage M2). First, even though immune infiltrates significantly correlate with ARL11 expression, the R values in Fig 6A are close to zero, and therefore an increase in ARL11 expression would result in a very small increase of immune infiltrates. Second, the data in Fig6B do not support that high ARL11 associates with shorter survival, they only support that high M2-TAMs associate with shorter survival. The significant effect is only due to the level of high M2-TAMs, which can be seen in the comparison shown in the figure between: 2 vs 1, which compares low ARL11 + high M2 (light blue line) with low ARL11 + low M2 (dark blue line), and 4 vs 3, which compares high ARL11 + high M2 (red line) with high ARL11 + low M2 (yellow line).

Response: Thanks a lot for your suggestion. Although we found the significant relevance between ARL11-related genes and immune response in our enrichment analysis, the correlation coefficients between ARL11 expression and immune infiltrates are less than +0.2. We speculate that ARL11 might regulate the immune processes through other unknown mechanisms, and we would like to validate the correlation between ARL11 expression and immune processes through biological experiments in future. Thus, we decide to remove the figure 6 in revised version of manuscript.

3. The expression values of the ARLs were evaluated in TIMER 2.0. Please include a sentence to inform the reader whether the results were the same in UALCAN and TIMER, i.e. higher in tumors than normal or vice versa. 

Response: Thanks for your careful review. We have compared the expression status of ARLs in UALCAN and TIMER databases and added related-paragraphs into the end of the Result 1 and added more expression details in S1 Table in revised version.

4. According to the manuscript, the methylation status of ARL4A is supposed to be seen in Table 1, but there is only ARL4C and ARL4D.

Response: Thanks for your careful review. In this study, we analyzed the relationship between the status of promoter DNA methylation with levels of ARLs mRNA expression. Among 22 ARLs, the correlation coefficient between mRNA expression and DNA methylation of ARL3, ARL4C, ARL4D and ARL11 were relatively higher (R ≥ 0.5, P < 0.05) . Therefore, we have shown the specific methylation site of ARL3, ARL4C, ARL4D and ARL11 in Table 1. In previous version of manuscript, we make some clerical errors, and we have corrected it in revised version.

5. „…..copy number amplification and DNA methylation might be involved in the epigenetic regulation of ARLs.“ Copy number amplification (CNA) is not considered an epigenetic mechanism. However, CNA may well regulate ARL expression levels.

Response: Thanks. Sorry for this inaccurate statement, we have corrected this point in revised manuscript.

6. The figure legends can be more detailed in many cases. For example, Fig 3 is completely missing a legend, Fig 4 would be easier to understand if more explanations were included, and in Fig 6 there is not a positive correlation with CD8+ T cells or M1-TAMs. Please amend.

Response: Thanks for your valuable suggestion. We have added more detailed information into figure legends.

7. They are called Circos plots with ´o´ not Circus with ´u´.

Response: Thanks for your suggestion. We have corrected these errors.

8. The text has been changed for the worse in the tracked version of the manuscript. The original text was more appropriate in many instances. Please fix.

Response: Thanks for your great advice. We have changed those inappropriate statements in revised version of manuscript.

Finally, we thank you and the reviewers again for your great efforts on improving the quality of this manuscript. We are looking forward to hearing your decision soon.

Thank you very much! 

Best wishes,

Yours sincerely,

Yahua Song

---

## [Decision Letter · Decision Letter 2]

3 Aug 2022

PONE-D-21-10417R2

ARL11 correlates with the immunosuppression and poor prognosis in breast cancer: a comprehensive bioinformatics analysis of ARL family members

PLOS ONE

Dear Dr. Song,

Thank you for submitting your manuscript to PLOS ONE. After careful consideration, we feel that it has merit but does not fully meet PLOS ONE’s publication criteria as it currently stands. Therefore, we invite you to submit a revised version of the manuscript that addresses the points raised during the review process.

Revision has improved the quality of manuscript. However, it needs English editing before its acceptance.

We look forward to receiving your revised manuscript.

Kind regards,

Chandi C. Mandal, Ph.D.

Academic Editor

PLOS ONE

Journal Requirements:

Reviewers' comments:

Reviewer's Responses to Questions

**Comments to the Author**

1. If the authors have adequately addressed your comments raised in a previous round of review and you feel that this manuscript is now acceptable for publication, you may indicate that here to bypass the “Comments to the Author” section, enter your conflict of interest statement in the “Confidential to Editor” section, and submit your "Accept" recommendation.

Reviewer #1: (No Response)

Reviewer #3: All comments have been addressed

2. Is the manuscript technically sound, and do the data support the conclusions?

Reviewer #1: Partly

Reviewer #3: Yes

3. Has the statistical analysis been performed appropriately and rigorously? 

Reviewer #1: Yes

Reviewer #3: Yes

4. Have the authors made all data underlying the findings in their manuscript fully available?

Reviewer #1: Yes

Reviewer #3: Yes

5. Is the manuscript presented in an intelligible fashion and written in standard English?

Reviewer #1: No

Reviewer #3: Yes

6. Review Comments to the Author

Reviewer #1: The revised manuscript is better than the previous one. The manuscript requires thorough reading be a native English Speaker. Authors need to attach a certificate that the manuscript has been checked by professional service provider.

Reviewer #3: Authors addressed all the response of both reviewers. The manuscript may be accepted for publication.

7. PLOS authors have the option to publish the peer review history of their article (what does this mean?). If published, this will include your full peer review and any attached files.

Reviewer #1: **Yes: **Ashok Kumar

Reviewer #3: **Yes: **Indranil Chattopadhyay

---

## [Author Response · Author response to Decision Letter 2]

23 Aug 2022

Dear editors,

We firstly thank all the reviewers for their thorough reading of our manuscript and for the constructive suggestions/criticisms.

In line with your suggestion, the revised version of manuscript has been performed professional English language editing and review services by AiMi Academic Services (www.aimieditor.com).

Finally, we thank you and the reviewers again for your great efforts on improving the quality of this manuscript. We are looking forward to hearing your decision soon.

Thank you very much! 

Best wishes,

Yours sincerely,

Yahua Song

---

## [Editor Report · Decision Letter 3]

6 Sep 2022

ARL11 correlates with the immunosuppression and poor prognosis in breast cancer: a comprehensive bioinformatics analysis of ARL family members

PONE-D-21-10417R3

Dear Dr. Song,

We’re pleased to inform you that your manuscript has been judged scientifically suitable for publication and will be formally accepted for publication once it meets all outstanding technical requirements.

Kind regards,

Chandi C. Mandal, Ph.D.

Academic Editor

PLOS ONE

Additional Editor Comments (optional):

Final revision has improved the quality of this manuscript and it is accepted.
---

## [Editor Report · Acceptance letter]

4 Nov 2022

PONE-D-21-10417R3 

ARL11 correlates with the immunosuppression and poor prognosis in breast cancer: a comprehensive bioinformatics analysis of ARL family members 

Dear Dr. Song:

I'm pleased to inform you that your manuscript has been deemed suitable for publication in PLOS ONE. Congratulations! Your manuscript is now with our production department. 

Kind regards, 

on behalf of

Dr. Chandi C. Mandal 

Academic Editor

PLOS ONE